META-RESEARCH

# The effect of the COVID-19 pandemic on the gender gap in research productivity within academia

**Abstract:** Using measures of research productivity to assess academic performance puts women at a disadvantage because gender roles and unconscious biases, operating both at home and in academia, can affect research productivity. The impact of the COVID-19 pandemic on research productivity has been the subject of a number of studies, including studies based on surveys and studies based on numbers of articles submitted to and/or published in journals. Here, we combine the results of 55 studies that compared the impact of the pandemic on the research productivity of men and women; 17 of the studies were based on surveys, 38 were based on article counts, and the total number of effect sizes was 130. We find that the gender gap in research productivity increased during the COVID-19 pandemic, with the largest changes occurring in the social sciences and medicine, and the changes in the biological sciences and TEMCP (technology, engineering, mathematics, chemistry and physics) being much smaller.

**KIRAN GL LEE\*, ADELE MENNERAT, DIETER LUKAS†, HANNAH L DUGDALE†, ANTICA CULINA\*†**

**\*For correspondence:**
kgllee1@sheffield.ac.uk (KGLL);
aculina@irb.hr (AC)

†These authors contributed
equally to this work

**Competing interest:** The authors
declare that no competing
interests exist.

**Reviewing Editor:** Peter
Rodgers, eLife, United Kingdom

## Introduction

Research productivity, defined as the number of manuscripts or publications, is a widely used, but flawed, metric for evaluating academic merit because it biases against individuals according to socio-demographic circumstances. Women are disadvantaged compared to men when success is measured using traditional metrics of research productivity (*Astegiano et al., 2019*; *Huang et al., 2020*), despite no actual differences in contribution and impact of research (*van den Besselaar and Sandström, 2016*; *van den Besselaar and Sandström, 2017*). Additionally, during the COVID-19 pandemic, novel living and working conditions worsened the research productivity of many women worldwide (*Anwer, 2020*; *Boncori, 2020*; *Guy and Arthur, 2020*; *Herman et al., 2021*; *Altan-Olcay and Bergeron, 2022*).

Multiple factors are likely to contribute to gendered changes in research productivity during a pandemic. First, women generally perform more unpaid caregiving and domestic work (*Schiebinger et al., 2008*; *Schiebinger and Gilmartin, 2010*). Social-distancing and facility closures during the pandemic increased caregiving and domestic work (*Carli, 2020*; *Carlson et al., 2020*) with reduced community help from nurseries, schools, care homes, house cleaners, laundrettes, nannies, babysitters and family (*Myers et al., 2020*; *Barber et al., 2021*; *Breuning et al., 2021*; *Deryugina et al., 2021*; *Shalaby et al., 2021*). As these tasks have disproportionately fallen on women, time and space for academic research during "work-from-home" conditions was difficult (*Abdellatif and Gatto, 2020*; *Boncori, 2020*; *Guy and Arthur, 2020*).

Second, the distribution of work within academic institutions is often gendered. Women undertake more 'non-promotable' tasks (*Babcock et al., 2022*) such as administrative, supportive and mentoring roles (*Porter, 2007*; *Mitchell and Hesli, 2013*; *Babcock et al., 2017*; *Guarino and Borden, 2017*; *O'Meara et al., 2017a*; *O'Meara et al., 2017b*). Changes in teaching and

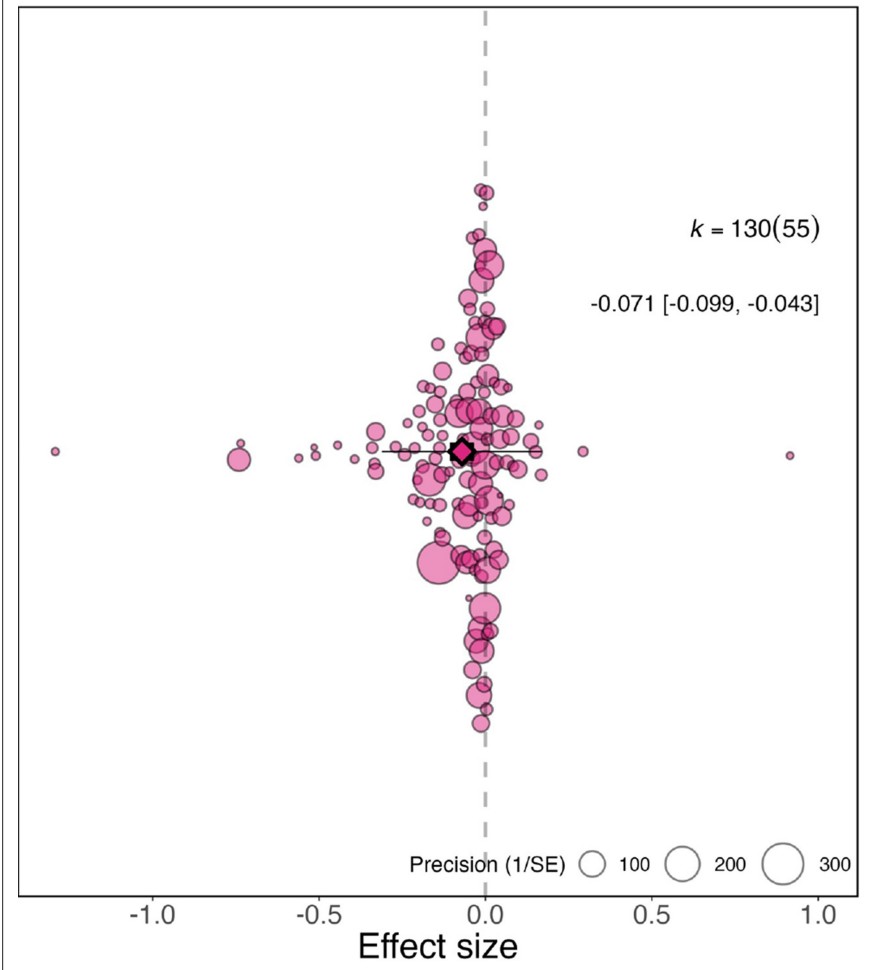

$k = 130(55)$

-0.071 [-0.099, -0.043]

**Figure 1.** Overall effect of the pandemic on the gender gap in research productivity. Orchard plot showing all 130 effect sizes (points), and the precision with which they were measured (point size). The plot shows the mean effect size (darker coloured point outlined in black and vertically centred), the 95% confidence interval (horizontal thick black bar), the 95% prediction interval of the expected spread of effect sizes based on between-study variance (horizontal thin black bar) and is centred at 0 (vertical dashed line). Points are spread vertically for presentation reasons to reduce overlap. $k$ is the total number of effect sizes; the 130 effect sizes shown here were calculated from 55 studies.

with evaluation being more influenced by cognitive shortcuts. These shortcuts are often associated with biases tending to operate against women (*Kaatz et al., 2014*; *Reuben et al., 2014*; *Carli, 2020*) resulting in lower success getting submissions accepted (*Fox and Paine, 2019*; *Murray et al., 2019*; *Day et al., 2020*; *Hagan et al., 2020*).

The role of these factors shaping the gender gap in research productivity during the pandemic might differ across research fields (*Madsen et al., 2022*). One possibility is that research fields that were already more gender-disparate may have experienced the most exacerbated gender gaps during the pandemic. In fields that were already traditionally more gender-disparate, less support may have been available to women to balance the effects of the pandemic. Male-dominated fields often lack viewpoints of female colleagues, and might therefore be less likely to identify and support paid care work or extended leave options (*Clark, 2020*; *Nash and Churchill, 2020*). An alternative possibility is that the pandemic might have eroded the support structures that existed in more gender balanced fields. The pandemic may also have exacerbated a gender gap in authorship position (first, middle or last) (*King and Frederickson, 2021*) if additional service, teaching, caregiving, and domestic roles taken up by female academics during the pandemic may limit their abilities to perform research (as first authors) or lead research (as last authors) but not in supporting research (as middle authors).

Here, we quantitatively calculated by meta-analysis the mean effect of the COVID-19 pandemic on the gender gap in research productivity and predicted the gap increased compared to the period just prior to the pandemic, such that male academic productivity saw even further increases. We assume that the pandemic might have influenced the multiple aspects that jointly affect gender inequality in research productivity, and our estimate reflects whether on average these effects have increased or decreased gender inequality.

First, as studies differ in the type of research productivity measured, between individual survey responses, numbers of submissions and numbers of publications, we investigated the influence this might have on the gender gap increase observed during the pandemic, but with no expectation of any differences.

Second, we explored variation in the gender gap increase across research fields and then explored the effect of research field according to the previous degree of gender disparity.

administration in response to the pandemic were therefore more likely to be facilitated by women (*Docka-Filipek and Stone, 2021*; *Minello et al., 2021*).

Third, labour roles contributing towards publication are also gendered. Women generally perform more technical work such as generating data, whilst men assume more core tasks in conceptualisation, analysis, writing and publishing (*Macaluso et al., 2016*). Pandemic closures to research institutions would therefore likely impact women authorship from technical roles stronger than men. Additionally, the surge in publications during the pandemic (*Else, 2020*) could have reduced the quality of peer review,

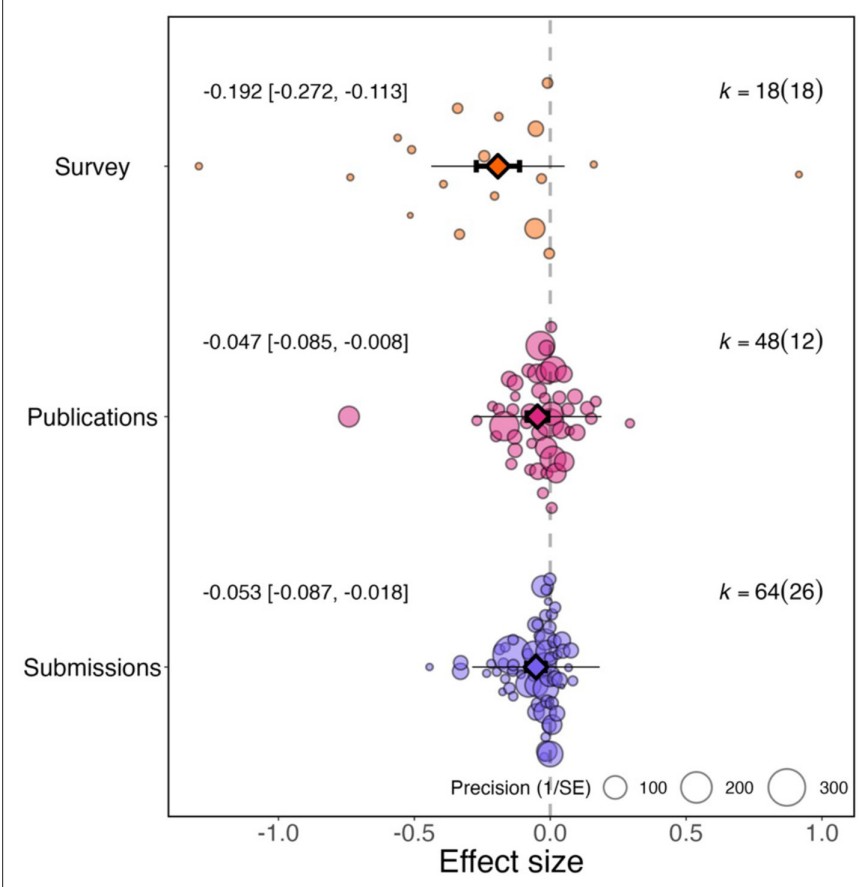

**Figure 2.** Effect of the pandemic on the gender gap in research productivity for three different measures of productivity. Orchard plots comparing the distribution of effect sizes (points) and the precision with which they were measured (point size) when the measure of research productivity is based on responses to surveys (top), number of publications (middle), and number of submissions (bottom). Each plot shows the mean effect size (darker coloured point outlined in black and vertically centred), the 95% confidence interval (horizontal thick black bar), the 95% prediction interval of the expected spread of effect sizes based on between-study variance (horizontal thin black bar) and is centred at 0 (vertical dashed line). Within each category, points are spread vertically for presentation reasons to reduce overlap. For each subgroup, *k* is the total number of effect sizes, and the number of studies from which these effect sizes were calculated is given inside the brackets.

We predicted the gender gap is exacerbated in fields that already had a previously greater gender gap, as according to the proportion of female authors, because of less support available to women to balance the effects of the pandemic.

Third, we explored whether the disparity in favourable authorship positions has increased. We predicted the gender gap has increased more in first and last, rather than middle authorship positions because female academics have been especially more limited in undertaking leading, but not supportive research roles in lockdown conditions.

## Results

Our systematic literature review identified 55 studies that met the inclusion criteria (for details on the procedure please see the Methods section). All of the identified studies only compared women to men (see Limitations). We extracted and calculated 130 effect sizes from these studies and performed a meta-analysis and meta-regression to test our three hypotheses and related predictions. Out of 130 effect sizes, 23 are based on survey responses (survey studies), and 107 are studies that measure the number of submitted or published articles (article studies).

### Has the pandemic increased the gender gap in research productivity?

Across the full dataset (N=130), after controlling for multiple effect sizes from the same study, we found the relative productivity of women to men decreased during the pandemic by −0.071 compared to before the pandemic (95% CI=−0.099 to −0.043, SE = 0.0144, p <0.001; *Figure 1*). This indicates that the relative productivity of women compared to men is 7% lower than what it was prior to the pandemic, meaning that in cases where men and women were estimated to be equally productive, the productivity of women now is only 93% that of men.

There is large variation in the 130 effect sizes, with 38 indicating a clear increase in the gender gap (95% confidence intervals within negative ranges) and 56 a trend of an increase (effect size is negative but 95% confidence intervals are not within negative ranges), while 11 indicate a clear decrease in the gender gap (95% confidence intervals within positive ranges) and 25 a trend of a decrease (effect size is positive but 95% confidence intervals are not within positive ranges). Total heterogeneity was high ($I^2$= 97.9%), with 46.6% of it explained by whether research productivity was measured by survey responses or submission/publication numbers and 52.1% explained by the individual effect sizes.

### Does the gender gap change depending on how it was measured?

The change in research productivity can be measured from survey responses (survey studies, N=23 effect sizes) or from the number of articles submitted or published (article studies, N=107). The degree of increase in the gender gap caused by the pandemic differed according to the type of research productivity measured (QM (df = 3)=37.130, P<0.001; *Figure 2*). Studies

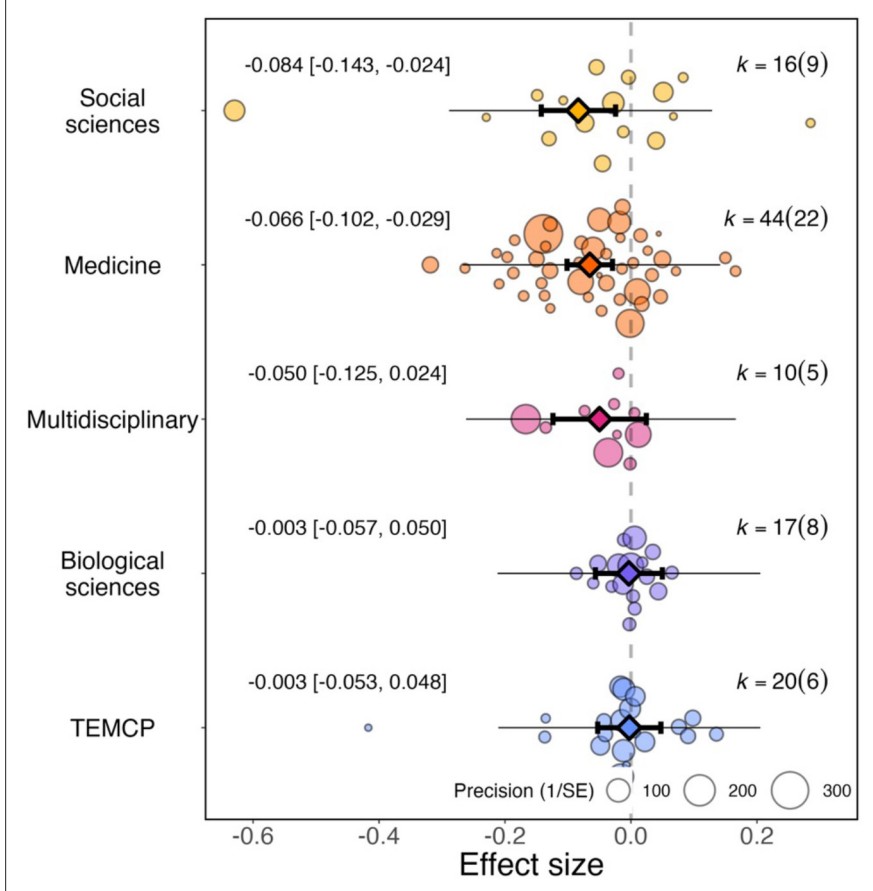

**Figure 3.** Overall effect of the pandemic on the gender gap in research productivity for five research fields. Orchard plot comparing the distribution of effect sizes (points) and the precision with which they were measured (point sizes) for five research fields. Each plot shows the mean effect size (darker coloured point outlined in black and vertically centred), the 95% confidence interval (horizontal thick black bar), the 95% prediction interval of the expected spread of effect sizes based on between-study variance (horizontal thin black bar) and is centred at 0 (vertical dashed line). TEMCP: technology, engineering, mathematics, chemistry and physics. Within each category, points are spread vertically for presentation reasons to reduce overlap. For each subgroup, *k* is the total number of effect sizes, and the number of studies from which these effect sizes were calculated is given inside the brackets.

*P*=0.001; *Figure 3*). When considering research fields individually, social sciences showed the greatest increases in the academic productivity gender gap during the pandemic (–0.084, 95% CI=−0.143 to −0.024, SE = 0.030, *P*=0.006), followed by medicine (–0.066, 95% CI=−0.102 to −0.029, SE = 0.019, *P*<0.001).

The pandemic showed little effect in multidisciplinary fields (–0.050, 95% CI=−0.125 to 0.024, SE = 0.038, *P*=0.188), biological sciences (–0.003, 95% CI=−0.057 to 0.050, SE = 0.027, *P*=0.902), or technology, engineering, mathematics, chemistry and physics (–0.003, 95% CI=−0.053 to 0.048, SE = 0.026, *P*=0.916).

### Has the pandemic exacerbated existing differences in gender disparity?

For the article studies with available data (N=99), we recorded the number of female and male authors before the pandemic, as defined by the time-period sampled in the respective study and used this ratio as a proxy for the size of the previous gender disparity in that population sampled. Based on this subset of data, we found that the pandemic has increased the gender gap in article output more in journals/repositories/pre-print servers that were previously less gender-disparate (QM(df = 1)=10.285, *P*=0.001).

When grouping studies by research fields (*Figure 4*), those with a smaller gender gap prior to the pandemic experienced greater gender disparity in academic productivity during the pandemic compared with fields where the gender gap was already large to start with (Social sciences: 35.8% to 33.4%, medicine: 36.6% to 33.8%, multidisciplinary: 36.2% to 34.2%, biological sciences: 32.7% to 32.7%, Technology, Engineering, Mathematics, Chemistry and Physics: 23.0% to 22.1%).

### Does the gender gap differ across authorship roles?

For article studies (N=107), we recorded whether first (N=54), middle (N=3), last (N=21), corresponding (N=15), or the total number of (N=14) authors were studied. Based on these data, we found no evidence of a significant differential impact of authorship position on effect sizes (QM(df = 5)=13.190, *P*=0.022; *Figure 5*).

The pandemic had a significant effect on first authorship roles (–0.040, 95% CI=−0.073 to −0.007, SE = 0.017, *P*=0.019) but not for all authorship roles (–0.045, 95% CI=−0.107 to 0.017, SE = 0.320, *P*=0.154), corresponding authorship roles (–0.058, 95% CI=−0.123 to

measuring changes to research productivity during the pandemic based on surveys detected a larger overall effect (–0.192, 95% CI=−0.272 to −0.113, SE = 0.041, *P*<0.001) than studies that compared the number of articles published (–0.047, 95% CI=−0.085 to −0.008, *P*=0.017, SE = 0.020) or submitted (–0.053, 95% CI=−0.087 to −0.018, *P*=0.003, SE = 0.017) by authors of each gender before and during the pandemic.

### Has the pandemic affected women differently across research fields?

For effect sizes from article studies grouped by research field (N=107), we found little evidence of a significant differential impact of research fields on the reported effect sizes (QM (df = 5)=21.967,

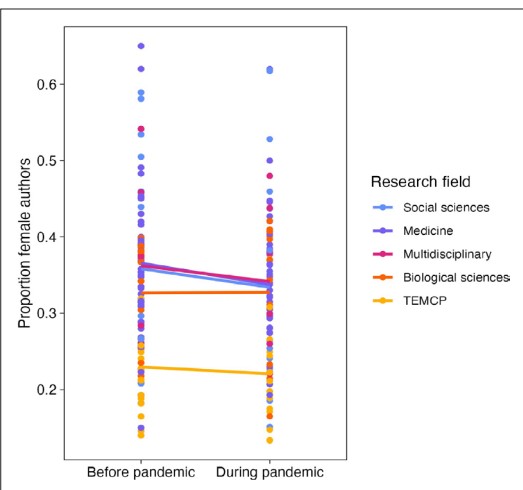

**Figure 4.** Effect of the pandemic on the gender gap in research productivity (as measured by number of articles submitted or published) for five research fields. Each point shows the proportion of female authors before (left) or during (right) the pandemic. The solid lines connect the mean value for each research field before and during the pandemic. The largest decreases are observed for the three fields that had the highest proportions of female authors (social sciences, medicine and multidisciplinary). TEMCP reflects the technology, engineering, mathematics, chemistry and physics fields, and multidisciplinary includes studies that span fields.

0.007, SE = 0.033, *P*=0.080), middle authorship roles (–0.045, 95% CI=−0.173 to 0.820, SE = 0.065, *P*=0.485) or last authorship roles (–0.040, 95% CI=−0.094 to 0.015, SE = 0.028, *P*=0.152).

### *Is there evidence of publication bias?*

The multilevel meta-regression, including article studies and survey studies as a moderator because of their differences in sample size, showed no evidence of publication bias (article studies: slope = –0.025, 95% CI=−0.059 to −0.009, SE = 0.017, *P*=0.148; survey studies: slope = –0.157, 95% CI=−0.244 to −0.071, SE = 0.044, *P*<0.001). This model correlates standard error with effect size and a negative slope suggests small studies do not have large effect sizes, with the negative slope among survey studies indicating the large heterogeneity that exists among these kinds of studies (see Discussion).

A visual inspection of the funnel plots similarly did not indicate any suggestion of publication bias (*Figure 6*).

### *Are our results robust?*

Our results changed little when we conducted a sensitivity analysis that excluded seven effect

sizes using four measures of productivity from survey studies that are less directly comparable: research time (N=4), job loss (N=1), burnout (N=1) and the number of projects (N=1). When excluding these effect sizes, the overall estimate was –0.063 (95% CI=−0.0892 to −0.0372, SE = 0.0133, *P*<0.001), and the estimate for survey studies only was –0.239 (95% CI=−0.3419 to −0.1352, SE = 0.0527, *P*<0.001).

Overly large effect sizes did not change the results in a leave-one-out analysis, which repeatedly fitted the overall model as in prediction 1 a, but for survey studies only, leaving out one effect size at a time to see the effect on the overall estimate for surveystudies (*Figure 6—figure supplement 1*). Leaving out the most influential effect size, the overall estimate was –0.183 (95% CI=−0.270 to −0.096, SE = 0.045, *P*<0.001).

## Discussion

Our study finds quantitative evidence, based on 55 studies and 130 effect sizes, to support the hypothesis that the COVID-19 pandemic has exacerbated gender gaps in academic productivity. These findings are consistent with the notion that novel social conditions induced by the pandemic have disadvantaged women in academia even more than before. Overall, the studies summarised in our meta-analysis suggest that gender gaps in research productivity within academia increased on average by 7% relative to the gender gaps that existed before the pandemic. We found no evidence of a publication bias in the studies investigating changes in the gender gap.

There is high heterogeneity in the effect sizes reported from different studies, arising from the type of research productivity measured. When measuring research productivity as the number of published or submitted articles, we find a slightly smaller increase in the gap of around 5%. This corresponds to the proportion of authors on submitted or published articles who are women declining from an average of 33.2% pre-pandemic to 31.4% during the pandemic (–0.05 * 33.2%=−1.7%). Such a change might reflect lower submission and acceptance rates of articles by women compared to their male colleagues or an increased drop-out of woman from academia caused by the pandemic. When measured by surveys, productivity reduction was 19% higher than in men. Future studies might therefore detect different effects of the pandemic on gender disparity in productivity, depending on the outcomes they assess and the timeframe over

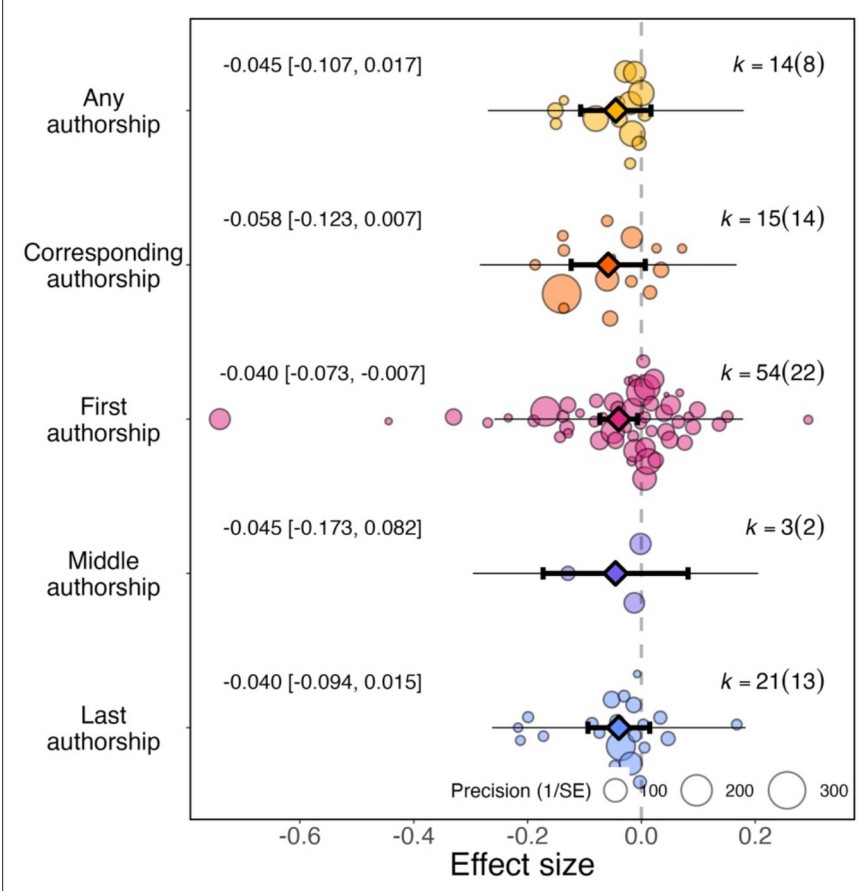

**Figure 5.** Effect of the pandemic on the gender gap in research productivity (as measured by number of articles submitted or published) for five authorship positions. Orchard plots comparing the distribution of effect sizes (points) and the precision with which they were measured (point sizes), for various authorship positions. Each plot shows the mean effect size (darker coloured points outlined in black and vertically centred), the 95% confidence interval (horizontal thick black bar), the 95% prediction interval of the expected spread of effect sizes based on between-study variance (horizontal thin black bar) and is centred at 0 (vertical dashed line). Within each category, points are spread vertically for presentation reasons to reduce overlap. For each subgroup, *k* is the total number of effect sizes, and the number of studies from which these effect sizes were calculated is given inside the brackets.

restricted access to laboratories, field sites and collaborators, many new projects have been delayed (*Corbera et al., 2020*). It is likely that the article studies we could include in our study underestimates the long-term effects of the pandemic, which might span over many years. However, the exact time dynamics are difficult to predict as adjustments and changes in conditions might lead to a normalising in production patterns over time (*Clark, 2023*). In support of this view, we find some indication for a larger, real-time effect from the effect sizes based on survey responses, which indicate a much stronger negative effect of the pandemic on women's productivity compared to men's (effect size = –0.192). This signals that women are nearly one fifth more likely than men to indicate that the pandemic has negatively affected their academic activities, which may stem from a combination of women on average feeling a larger strain, and a larger proportion of women being severely affected by the pandemic. In the literature used within our meta-analysis, five of six survey studies report evidence of a negative interaction effect of being both female and a parent on research productivity during the pandemic, presumably because of increased caregiving demands.

Our analysis suggests the pandemic may have differentially impacted female researchers across research fields, with increases in gender gaps particularly visible in research fields that were nearest to being gender-equal before the pandemic. Social sciences and medicine were two fields closest to gender equality that experienced the most significant decrease in female authors. Female researchers working in fields with previously gender-equitable environments may have experienced new, difficult research conditions induced by the pandemic, whereas in gender-biased fields, these difficulties might already have been present. Alternatively, social sciences and medicine are fields that could have had the greatest surge in COVID-19 and pandemic-related research. Women in social sciences and medicine potentially had less opportunities to pursue this new pandemic-related research because of extra work performed in gender roles, or because women already had relatively smaller collaborative networks, fewer senior positions, and less funding. Additionally, many medical journals sped up the publication process (*Horbach, 2020*), so the real-time effect of the pandemic on research productivity in women versus men may be reflected more in papers submitted and published in medicine than in other fields.

which these changes are analysed. As our data and analytical codes are open, and the literature on pandemic effects is increasing, we hope our work can form a first step in a living systematic review on the topic.

Our study likely underestimates the pandemic effect on article productivity in women because writing and publishing can take a long time (*Powell, 2016*). Many of the articles submitted or published during the pandemic were likely started and at least partially completed prior to the pandemic, given that most research grants span multiple years. Most of the studies in our sample obtained their data relatively soon after when the WHO officially declared a pandemic (median of 7 months after January 2020). With

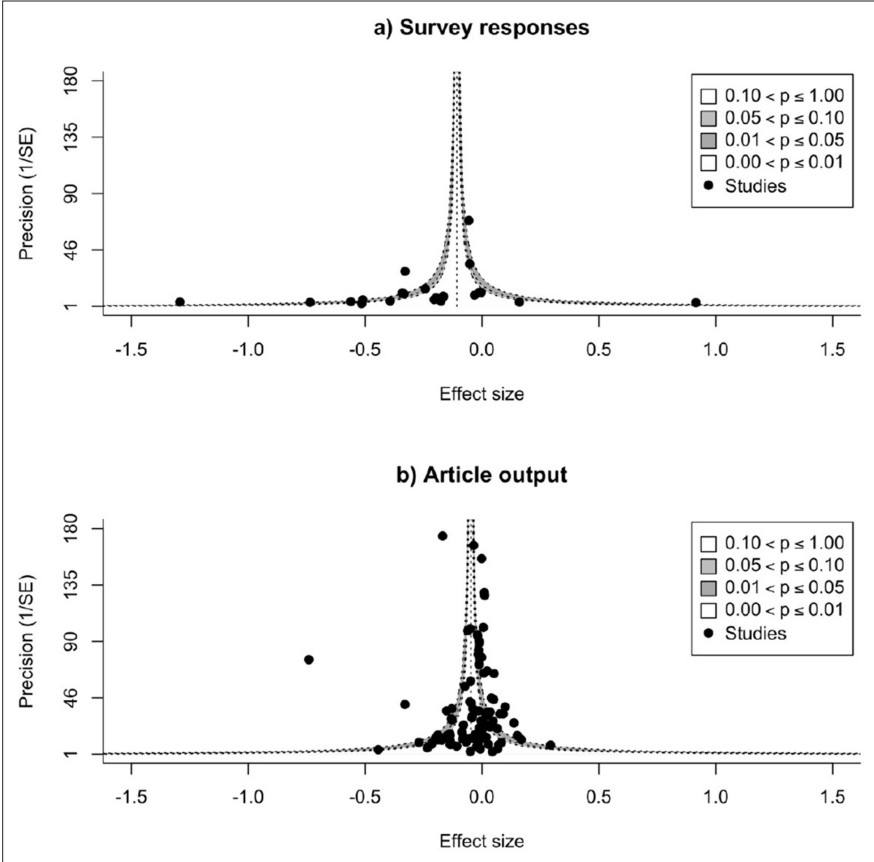

**Figure 6.** Checking for possible publication bias. Funnel plots of effect sizes and their precision, as a function of standard error, for studies that measure research productivity by responses to a survey (**A**), and by number of articles published (**B**). The vertical dashed line is the summary effect size; the legend outlines levels of statistical significance for effect sizes based on their precision. Neither plot shows evidence for publication bias.

The online version of this article includes the following figure supplement(s) for figure 6:

**Figure supplement 1.** Forest plot of leave-one-out analysis for survey studies only.

We found that the pandemic increased the gender gap particularly among first authors, potentially suggesting that women were particularly more restricted in the time they had available to write papers. However, we cannot exclude that other authorship positions underwent a similar increase in the gender gap because the samples were uneven, with half of effect sizes focussed on first authorship roles. Additionally, not all fields have the same authorship order norms making comparisons difficult.

It seems unlikely that this change in the gender gap during the pandemic simply represents a normal temporal fluctuation. The survey results, which report the strongest effects, specifically focused on the influence of the pandemic above and beyond the pressures researchers might already normally experience. The 5% decline in the proportion of authors who are women also

likely indicates the extraordinary circumstances of the pandemic. This decline is steep relative to a study comparing the change in the proportion of female authors between 1945 and 2005 that showed a steady increase from 14% of all authors being women to 35%, with no apparent year-on-year decline since at least 1990 (*Huang et al., 2020*). Other studies that explore mechanisms affecting gender equality during the pandemic may offer insight into the trend of reduced relative research productivity of women to men that we find in this study. These mechanisms are likely to be a combination of gender inequalities that affect all women during the pandemic such as changes in carer roles and financial stability (*Collins et al., 2021*; *Fisher and Ryan, 2021*; *Flor et al., 2022*), and those specifically affecting women working in academia such as changes in the potential to start new projects (*Malisch et al., 2020*; *Herman et al., 2021*; *Pereira, 2021*) and changes in research topics and publication processes to cover new topics (*Viglione, 2020*; *Clark, 2023*).

Our study has several limitations, which are fully outlined in the Limitations section of the methods and summarised here. Causes of the increased gender disparity in research productivity during the pandemic are not identified. Gender is investigated only as a binary variable, mostly using first-name prediction tools, with no investigation into non-binary or transgender. Geographic regions are not investigated, limiting generalisability as our samples are not equally representative of all geographic regions. All studies are to some extent unique by combined differences in sampling method and analyses performed. Survey studies may unintentionally sample respondents with extreme opinions and can suffer from recall limitations and self-serving bias. This diversity in approaches does however offer future studies the potential to investigate which specific mechanisms might have interacted to shape the overall increase in the gender gap we observe.

## Conclusion

Overall, our study highlights exacerbated gender gaps in academic research productivity during the COVID-19 pandemic. Despite the heterogeneity in our sample with regards to different outcome measures reflecting research productivity and different approaches to estimate effects, we overall find that most studies performed to date indicate that the gender gap in research productivity has increased during the pandemic. This

finding suggests that the COVID-19 pandemic has likely influenced many of the processes that contribute to differences in the achieved research productivity of women compared to men. This gender gap was exacerbated more in social sciences and medicine, which are fields that were previously less gender-disparate and may represent regression in progress made towards gender equality.

Our study cannot speak to the potential mechanisms that might have led to changes in gender inequality during the pandemic and is therefore limited in deriving suggestions for potential interventions to potentially ameliorate unfair differential productivity besides indicating that such inequalities appear pervasive. Academic institutions should acknowledge and carefully accommodate the pandemic period when using research productivity to evaluate female academics for career progression in the coming years. For example, tenure-clock extensions designed to accommodate pandemic disruption may inadvertently exacerbate the gender gap by extending the period that advantaged individuals can outperform. More emphasis must be placed evaluating academic merit using more holistic measures and on an individual basis. We recommend future studies to investigate potential mechanisms of the increased gender gap in academic productivity, to continue monitoring the gendered disparity of academic productivity, and to investigate any long-term implications that can arise from reduced productivity of women.

## Methods

### Search process

We carried out a systematic review to identify, select and critically evaluate relevant research through data collection and analysis. We reported it following PRISMA guidelines (*Moher et al., 2009*). We carried out the literature search process in three steps: (1) a scoping search, (2) an initial search with pre-selected author terms, and (3) a refined search using terms as recommended by the litsearchR 1.0.0 (*Grames et al., 2019*). All searches were filtered for texts from 2020 onwards. We initially performed a scoping search to determine if there were over ten texts with primary research investigating differences by gender in academic productivity before and during the pandemic. The scoping search was conducted on 30/06/2021 by Google searching combinations of synonyms for: (1) the COVID-19 pandemic, (2) gender, (3) academia, (4) inequality

and (5) productivity. The scoping search identified 21 original research publications with quantitative metrics investigating differences in academic productivity by gender before and during the pandemic (*Supplementary file 1*, "Scoping search" sheet). Of these 21 studies (scoped texts), 14 were indexed by Web of Science, and 17 (including the same 14 from Web of Science) were indexed by Scopus.

Terms for the initial search were selected by scanning the title, abstract and keywords of scoped texts. We constructed an initial Boolean search string according to the PICO (Population, Intervention, Comparator, Outcome) framework (*Livoreil et al., 2017*). Population was represented by "academia", Intervention by "pandemic", Comparator by "gender" and Outcome by "inequality" and "productivity" (see Table S1 in *Supplementary file 2*). A sixth concept group contained terms used to exclude irrelevant studies that did not investigate studies in hypothesis one. Terms within concept groups were connected by the Boolean OR operator, and the concept groups were connected by the AND or AND NOT operators, enabling searches for any combination that includes one term from each of the six concept groups. Terms in the initial search were selected by scanning the title, abstract and keywords of scoped texts. The initial search in Scopus generated 722 texts, including 13/17 (76.5%) of scoped texts indexed by Scopus.

To improve the 13/722 (1.8%) efficiency of finding scoped texts from our initial search, we imported all 722 texts into R and used litsearchR. Using litsearchR, potential key terms were extracted from the title, abstract and keywords of texts using the Rapid Automatic Keyword Extraction algorithm. A ranked list of important terms was then created from building a key term co-occurrence network (see Table S2 in *Supplementary file 2*). Six high-strength terms within the key term co-occurrence matrix, describing research not relevant to our study, such as those of an epidemiological or experimental nature, were added to the AND NOT operator concept group to exclude texts mentioning these terms. *Table 1* describes terms of the refined Boolean search string and their respective concept groups. We performed the refined search on 27/07/2021 and generated 700 total texts combined from Scopus (126 texts, including 13/17 studies found in the scoping search), the Web of Science core collection (199 texts), EBSCO (276 texts and Proquest (99 texts)) from 2020 onwards. The final search hit rate had an efficiency of 10.3% (13/126) on Scopus. After removing duplicates, 580 texts

**Table 1.** Final Boolean search string used in full literature search for texts since 2020. Terms in italics were added using litsearchR.

| Concept group | PICO group | Terms |
|---|---|---|
| Academia | Population | (academi* OR author* OR database* OR journal* OR research OR scien*) |
| Gender | Population | AND (female* OR gender OR male* OR men OR women) |
| Pandemic | Intervention | AND (coronavirus OR covid OR pandemic) |
| Inequality | Comparator | AND (bias* OR disparit* OR disproportion* OR fewer OR gap OR "gender difference*" OR imbalance* OR inequalit* OR inequit* OR parity OR "sex difference*" OR skew* OR unequal) |
| Productivity | Outcome | AND (performan* OR publication* OR publish* OR productiv*) |
| Exclusion of biomedical studies | Population | AND NOT (experiment OR laboratory OR mortality OR surviv* OR *"acute respiratory"* OR *gis* OR *icu* OR *risk* OR *rna* OR *symptoms*) |

remained to enter the study screening stage. We did not perform the search in any grey literature databases.

### Study screening

To be included in our meta-analysis, studies had to quantitatively investigate gender differences in productivity within academia before and during the pandemic. In our initial screen of titles, abstracts and keywords of studies we kept only those suggesting the study investigated: (1) academia, (2) gender, (3) the COVID-19 pandemic and (4) some measure of productivity (see Table S3 in *Supplementary file 2*). We included any text returned by the database, of any publication status, including grey literature in any language, though all texts had abstracts with an English version. To ensure repeatability of the screening process, we used Rayyan.ai (*Ouzzani et al., 2016*) to blind the inclusion or exclusion of 420 randomly selected studies by two reviewers (KGLL and DL). The agreement rate between reviewers was 97%, with 49 studies that both authors agreed to include, 357 studies which both excluded, ten studies one reviewer included but the other excluded, and four studies only included by the other reviewer. This agreement rate resulted in a, "strong" (*McHugh, 2012*) to "near perfect" (*Landis and Koch, 1977*) Cohen's kappa of 0.86. Of the 14 studies which

were included by one but excluded the other, three were included after joint review. To reduce workload, we did not double-initial-screen 160 remaining texts because of 97% agreement rates achieved during the sample of double-screened texts. Overall, out of the 580 texts, 70 were retained for the full text screening (*Supplementary file 3*).

Full texts were then screened to identify articles that had included four qualifiers: (1) both genders, (2) some quantifiable metric of academic productivity measured, and (3) compared a time period before the pandemic with a time period during the pandemic (time periods chosen according to the authors' discretion), (4) primary data. Full-text screening was conducted by recording which, if any of the four qualifiers were missing for each text in an excel spreadsheet and only performed by KGLL. Studies were not assigned quality weightings as there is no common standard to do this. Thus, 25 studies that all contained necessary metrics to calculate effect sizes were retained for data extraction, excluding 45 studies (*Supplementary file 3*).

### Iterating the search

To find studies that had been published since the 27/07/21 search (*Table 1*), we iterated the search and screen process. The second search was repeated on 28/02/2022, generating 1646 total

texts combined from Scopus (258 texts, including 14/17 studies found in the scoping search), the Web of Science core collection (413 texts), EBSCO (542 texts) and Proquest (433 texts) from 2020 onwards. We removed 438 duplicates using Rayyan.ai, leaving 1208 de-duplicated studies. To ensure our methods are repeatable, we checked and found all 580 de-duplicated studies from the previous search were also found again. Out of the 1208 texts from the final search, we included 170 after screening titles, abstracts and keywords. For these 169, we screened the full texts, excluding 120 studies (53 studies missed one qualifier, 38 studies missed two qualifiers, 27 studies miss three qualifiers, 1 study missed four qualifiers) and keeping 50 (including the 25 identified in the original search) that all contained the necessary information to calculate the effect sizes (*Supplementary file 3*). Two studies with full texts in Spanish were translated to English by Google translate, of which one was included. Five studies found in the scoping search but not returned in database searches were also included, resulting in 55 total studies used to extract variables for analysis. The full PRISMA flow diagram outlining the number of texts included at each stage in first and iterated search is found in Figure S1 in *Supplementary file 2*.

### Our sample

Our sample consists of 130 effect sizes obtained from 55 studies including surveys of potentially affected people (*Myers et al., 2020*; *Rodríguez-Rivero et al., 2020*; *Barber et al., 2021*; *Breuning et al., 2021*; *Camerlink et al., 2021*; *Candido, 2021*; *Deryugina et al., 2021*; *Ovalle Diaz et al., 2021*; *Ellinas et al., 2022*; *Gao et al., 2021*; *Ghaffarizadeh et al., 2021*; *Guintivano et al., 2021*; *Hoggarth et al., 2021*; *Krukowski et al., 2021*; *Maguire et al., 2021*; *Plaunova et al., 2021*; *Shalaby et al., 2021*; *Staniscuaski et al., 2021*; *Yildirim and Eslen-Ziya, 2021*; *Davis et al., 2022*; *Stenson et al., 2022*), and comparisons of numbers of articles submitted or published by gender before and during the pandemic (*Amano-Patiño, 2020*; *Andersen et al., 2020*; *Bell and Green, 2020*; *Cushman, 2020*; *Inno et al., 2020*; *Kibbe, 2020*; *Vincent-Lamarre et al., 2020*; *Wehner et al., 2020*; *Bell and Fong, 2021*; *Biondi et al., 2021*; *Cook et al., 2021*; *DeFilippis et al., 2021*; *Forti et al., 2021*; *Fox and Meyer, 2021*; *Gayet-Ageron et al., 2021*; *Gerding et al., 2021*; *Ipe et al., 2021*; *Jemielniak et al., 2021*; *Jordan and Carlezon, 2021*; *King and Frederickson, 2021*;

*Lerchenmüller et al., 2021*; *Mogensen et al., 2021*; *Muric et al., 2021*; *Nguyen et al., 2021*; *Quak et al., 2021*; *Ribarovska et al., 2021*; *Squazzoni et al., 2021*; *Williams et al., 2021*; *Anabaraonye et al., 2022*; *Ayyala and Trout, 2022*; *Chen and Seto, 2022*; *Cui et al., 2022*; *Harris et al., 2022*; *Wooden and Hanson, 2022*).

### Extracting variables

#### Effect size

We extracted values needed to calculate 130 effect sizes from 55 articles investigating the effect of the pandemic on academic research productivity of both genders, comparing the productivity before and during the pandemic, using time periods chosen according to authors' discretion. We calculated our own effect sizes wherever possible using the available summary statistics and/or statistical inferences. For 10 effect sizes which did not have data available to calculate our own effect sizes, we used already calculated percentage changes in the gender gap in academic productivity as predicted from lasso regression (N=2), Somers' delta (N=2), ordered logistic regression (N=1) and mixed-effect models (N=5). For 120 effect sizes, we entered summary data (N=117) or simple statistical tests (N=3) into Campbell collaboration's effect size calculator (*Wilson, 2019*) to calculate a standardised mean difference (d) effect size. For effect sizes calculated using summary data, 99 relied on the proportion of raw numbers of female and male authors before and after the pandemic, and 18 on the mean changes and standard deviations or standard errors in research productivity changes during the pandemic for female and male researchers. For effect sizes calculated from reported simple statistical tests, one converted the f-test statistic and sample size from a general linear model investigating the effect of gender on perceived work production, one converted the chi-square comparing proportions of female and male academics that experienced productivity changes due to the pandemic, and one converted the p-values from a t-test comparing mean changes in research time due to the pandemic. 10 effect sizes were calculated by obtaining raw numbers from graphs, estimated using Adobe Acrobat's measure tool (*Supplementary file 1*, "Calculations" sheet). Two effect sizes (*Jemielniak et al., 2021*; *Stenson et al., 2022*) were calculated using sample sizes obtained by personal correspondence with the article authors. Six studies investigated numbers of articles at different time points during the course of the

pandemic. From these studies, we calculated 30 effect sizes using numbers of articles across the entire pandemic period. We calculated multiple effect sizes from one study if they were referring to different research fields or authorship positions. We set the sign for effect sizes as negative if the pandemic had reduced relative research productivity of women (increased gender gap) and positive if the pandemic had increased the relative research productivity of women (reduced gender gap). A subset of 59 effect sizes were double-checked by A.C., A.M. and D.L and inconsistencies were discussed to ensure repeatability. K.L. then extracted the remaining 71 effect sizes.

### Variance
Of 10 effect sizes already calculated in the original studies, 7 provided variance as the standard error, which we squared to obtain the variance; and 3 provided the variance as 95% confidence intervals, which we divided by 1.96 and then squared (*Nakagawa et al., 2022*). For the other 120 effect sizes, variance was estimated in the Campbell collaboration calculator (*Wilson, 2019*) when calculating effect sizes.

### Research productivity measure
We first recorded whether the change in research productivity was measured from survey responses (survey studies, N=23 effect sizes) or from the number of articles submitted or published (article studies, N=107 effect sizes). Survey studies measured change in research productivity during the pandemic for each gender based on academics self-reporting their gender and change in general productivity (N=11 effect sizes), number of submissions (N=5 effect sizes), research time (N=4 effect sizes), number of projects (N=1 effect sizes), burn-out (N=1 effect sizes), or job loss (N=1 effect sizes). As 5 survey studies measured research productivity in the number of submissions, we included these studies in the articles submitted and published category. This resulted in 18 effect sizes from surveys measuring some aspect of research productivity, 64 effect sizes measuring numbers of article submissions, and 48 effect sizes measuring numbers of publications.

### Research field
For the article studies (N=107 effect sizes), we recorded the research field sampled based on the description in the original studies as either Medicine (N=44 effect sizes), Technology, Engineering, Mathematics, Chemistry and Physics

(N=20 effect sizes), Social sciences (N=16 effect sizes), Biological sciences (N=17 effect sizes), or Multidisciplinary (N=10 effect sizes), following the classification scheme of *Astegiano et al., 2019*.

### Previous gender disparity
For the article studies with available data (N=99 effect sizes), we recorded the number of female and male authors before the pandemic, as defined by the time-period sampled in the respective study and use this ratio as a proxy for size of the previous gender disparity in that population sampled.

### Authorship position
For the article studies (N=107 effect sizes), we recorded whether first (N=54 effect sizes), middle (N=3 effect sizes), last (N=21 effect sizes), corresponding (N=15 effect sizes), or any (N=14 effect sizes) authorship positions were studied. We classified one effect size studying submitting authors, as studying corresponding authors (*Fox et al., 2016*) and two effect sizes studying sole authors as studying last authors (*Moore and Griffin, 2006*).

We also extracted data for the following variables to enable description of the datasets: timeframe before the pandemic; timeframe during the pandemic; geographic region; data availability for gender and geographic region interaction effect; data availability for gender and career stage/age interaction effect; data availability for gender and parent status interaction effect; gender assignment accuracy threshold for article studies and gender inference method used. Please see *Supplementary file 1*, "Variables" sheet, for descriptions of all the variables.

### *Analyses*
We conducted all analyses in R 3.6.2 (*R Development Core Team, 2022*). We used the 'metafor' package 3.0.2 to fit models, and build funnel and forest plots (*Viechtbauer, 2010*). We used 'orchaRd' 0.0.0.9000 to build orchard plots to visualise distribution of effect sizes (points) and their precision (point size), calculated as a function of standard error (*Nakagawa et al., 2021*).

We fitted separate models for each prediction. All models included the identity of the article the effect size was extracted from as a random effect to control for dependency in effect sizes obtained from the same study. Models that include moderators use

an omnibus test of parameters reported as a QM metric, which tests the null hypothesis that all moderator effect sizes are equal and is significant when at least two moderators are different. We tested prediction 1 a in a model investigating the overall effect size and we displayed this as an orchard plot. We then tested prediction 1b in a model investigating the method of measuring research productivity (survey responses, number of submissions and number of publications) as a moderator of effect size and displayed this as an orchard plot. We included the outlier (*Jemielniak et al., 2021*) in the funnel plot of article studies because this effect size was obtained by personal correspondence clarifying the sample sizes used in the study, which we assume was verified. We tested prediction 2 a in a model investigating research field as a moderator of effect size for article studies in a model and displayed this as an orchard plot. We tested prediction 2b in a model investigating how previous gender disparity in research productivity before the pandemic, as measured by the proportion of female authors, influenced effect size and displayed this as a line graph, grouped by research field. To test prediction 3 a, we tested in a model authorship position as a moderator on effect size for publication studies. We tested for publication bias by performing a multilevel regression model (*Nakagawa et al., 2022*) which investigates whether small studies have large effect sizes, including research productivity measure as a moderator because of differences in sample size between article studies and survey studies. We display this relationship in funnel plots. We tested for total heterogeneity ($I^2$) using the 'i2_ml' function in 'orchaRd'. We applied a sensitivity analysis testing prediction 1 a (overall pandemic effect on gender gap) and prediction 1b (method of research productivity effect on pandemic gender gap) excluding seven effect sizes using four measures of productivity from survey-based studies that are less directly comparable: research time (N=4), job-loss (N=1), burnout (N=1) and the number of projects (N=1). We also performed a leave-one-out analysis using the 'leave1out' function in 'metafor'. This performed a meta-analysis on survey studies, leaving out exactly one study at a time to see the effect of individual studies on the overall estimate for survey studies. A full PRISMA checklist is found in Table S4 in *Supplementary file 2*.

## Limitations

Our focus is on comparing the effect of the pandemic on women relative to men. We recognize that gender extends beyond this comparison, and that biases are even more likely to target individuals whose identities are less represented and often ignored. These biases also reflect in a lack of studies of the full diversity of gender. While several of the surveys we include had the option for respondents to identify beyond the binary women/men, none of these studies report on these individuals, presumably because of the respective small samples. In addition, studies using numbers of submissions or publications (38 out of 55) to measure research productivity used automatic approaches that are more likely to mis-gender individuals as they inferred binary gender based on first names. While these approaches seemingly offer the potential to identify trends in larger samples, they themselves introduce and reinforce biases in relation to gender that are hard to assess, intersecting with biases in ethnicity as these approaches are often restricted to names common in English speaking countries (*Mihaljević et al., 2019*). For survey studies, only 18 effect sizes were used. These had a large heterogeneity in effect sizes, possibly reflecting subtle differences in the measure of research productivity asked in the survey. Surveys sample limited numbers of respondents, potentially biased towards sampling those holding extreme opinions of the pandemic. Subjectivity in survey responses could skew the estimate because of recall limitations and self-serving bias. We do not include grey literature databases in our searches, which may bias our samples to studies with positive effects. We did not perform forwards or backwards searches, meaning we may have missed some relevant studies. However, we expect the literature on the topic to grow, and hope that further work will build on our study and add these new effect sizes to our dataset. Most studies explored academic populations worldwide (N=99 effect sizes), or from Western (N=28 effect sizes) regions, but not the Global South (N=3 effect sizes) limiting investigation of interaction effects between geographic regions. Although 22/130 effect sizes from 8/55 studies held data subdivided between geographic regions, we did not extract separate effect sizes as they differed in the scale

of geographic region sampled, which limited our ability to make geographic comparisons. Conclusions from survey studies are also limited to North American and Western European, since 18/23 studies are exclusive to or have the majority of respondents from these regions. We recognise there are differences between article- studies in the length of time considered as before the pandemic (mean = 11 months, standard deviation = 10 months, range = 1–50 months) and during the pandemic (mean = 7 months, standard deviation = 5 months, range = 1–17 months). Survey studies were fielded at different times, (mean = 21/08/2020, standard deviation = 99 days, range = 20/04/2020 – 28/02/21) which potentially affects participants' beliefs of productivity changes. Investigating research field and authorship position effects is limited by the unequal and sometimes small sample sizes of variables that are compared. We used raw data to calculate effect sizes using the same modelling techniques wherever possible. This was not possible in 10 studies, where we consequently used effect size as provided in the study. We recognize that their different modelling techniques may have contributed to the estimated effect sizes. The patterns we describe should be seen as a potential indication that biases exist, but alternative approaches are needed to speculate about potential underlying causes and remedies.

### Acknowledgements

We express gratitude for papers not behind paywalls.

**Kiran GL Lee** is at the Groningen Institute for Evolutionary Life Sciences, University of Groningen, Groningen, The Netherlands, and the Department of Animal and Plant Sciences, University of Sheffield, Sheffield, UK
kgllee1@sheffield.ac.uk
https://orcid.org/0000-0003-1139-4853
**Adele Mennerat** is in the Department of Biological Sciences, University of Bergen, Bergen, Norway
https://orcid.org/0000-0003-0368-7197
**Dieter Lukas** in the Department of Human Behavior, Ecology and Culture, Max Planck Institute for Evolutionary Anthropology, Leipzig, Germany
https://orcid.org/0000-0002-7141-3545
**Hannah L Dugdale** is at the Groningen Institute for Evolutionary Life Sciences, University of Groningen, Groningen, The Netherlands
https://orcid.org/0000-0001-8769-0099

**Antica Culina** is at the Rudjer Boskovic Institute, Zagreb, Croatia, and the Netherlands Institute of Ecology, NIOO-KNAW, Wageningen, The Netherlands
aculina@irb.hr
https://orcid.org/0000-0003-2910-8085

*Author contributions:* Kiran GL Lee, Conceptualization, Resources, Data curation, Software, Formal analysis, Investigation, Visualization, Methodology, Writing – original draft, Project administration, Writing – review and editing; Adele Mennerat, Conceptualization, Data curation, Supervision, Validation, Methodology, Project administration, Writing – review and editing; Dieter Lukas, Conceptualization, Data curation, Formal analysis, Supervision, Validation, Methodology, Writing – review and editing; Hannah L Dugdale, Conceptualization, Data curation, Formal analysis, Supervision, Validation, Methodology, Writing – review and editing; Antica Culina, Conceptualization, Data curation, Formal analysis, Supervision, Validation, Methodology, Project administration, Writing – review and editing

*Competing interests:* The authors declare that no competing interests exist.

### Funding

No external funding was received for this work.

**Decision letter and Author response**
Decision letter https://doi.org/10.7554/eLife.85427.sa1
Author response https://doi.org/10.7554/eLife.85427.sa2

## Additional files

### Supplementary files
• Supplementary file 1. Scoping search, calculations and variables.

• Supplementary file 2. Tables S1–S4 and Figure S1.

• Supplementary file 3. Study screening.

• MDAR checklist

### Data availability
All data and materials to reproduce the meta-analysis can be found at Zenodo: https://doi.org/10.5281/zenodo.8116754.

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
