## [Decision Letter]

**Decision letter after peer review:**

Thank you for submitting your article "Meta-research: The effect of the COVID-19 pandemic on the gender gap in research productivity within academia" to *eLife* for consideration as a Feature Article. Your article has been reviewed by two peer reviewers, and the evaluation has been overseen by a member of the *eLife* Features Team (Peter Rodgers). The following individuals involved in review of your submission have agreed to reveal their identity: Emil Bargmann Madsen.

The reviewers and editors have discussed the reviews and we have drafted this decision letter to help you prepare a revised submission. The points that need to be addressed in a revised version are listed below. As you will see, there are quite a few points that need to be addressed.

Summary

This study provides fundamental evidence substantially advancing our understanding of the gender-specific impacts of the Covid-19 pandemic on academic research productivity. However, there are a number of concerns that need to be addressed.

Essential revisions

1) I am concerned about the use of the included studies for a formal meta-analysis. Firstly, it is clear from the authors well-structured description in the methods section that many studies are not measuring the same outcome.

Many studies use number of submitted or published articles, which may be a fairly comparable metric, but many of the survey studies use very different measures of productivity:

a. Research time (N = 4), job-loss (N = 1), burnout (N = 1), and number of projects (N = 1).

b. These are highly different outcomes, and should not be analysed as comparable.

2) Secondly, it is not clear how the studies differ in their use of the pandemic as an "intervention".

a. Some studies will ask respondents to compare their productivity before and after, while some studies will provide actual comparisons of productivity before and after. Both methods will have their share of drawbacks, but I do not think they are comparable.

b. The different survey studies will also have been fielded at different times, making the "intervention" (the pandemic) very different. Perhaps the loss of research time is most acute in the beginning, before academics have had time to adapt? Or later, because of burn-out?

3) Thirdly, the diversity of research design and modelling techniques makes it hard to conduct formal comparisons of effect estimates.

None of the included studies are randomised controlled experiments. In fact, the research design (and modelling techniques) differs immensely across included studies. Therefore:

a. We do not know to what degree individual, and thus by extension, individual gender gap estimates can be causally interpreted. This should be more explicitly acknowledged in the article.

b. Different estimates may be relatively closer to a causal effect, depending on the research design. By treating all designs equally, the authors gloss over these crucial differences in what we can gleam from each study.

c. It is not clear what "estimand" we get from the meta-analytical model. Some studies are cross-sectional, and will give us the average between-subject difference in productivity. Other studies, longitudinal ones, may give us the "difference in differences", i.e. the average difference in within-subject publication productivity trend. These are fundamentally two different answers to the question: Has the pandemic exacerbated the gender gap in research?

d. The guidance from e.g. Cochrane is to use a random effects model to account for variation in what effect is estimated. The authors seem to use a random effects model to account for dependency between effects extracted, but I think they should also be more explicit in leveraging this for accounting for variation in what effect is estimated.

e. Each included paper will have different modelling strategies. E.g. the Myers et al. 2020 paper uses regularised regression (Lasso), while the Andersen et al. (2020) used a hierarchical logit model. These differences may contribute to differences in estimated gender gaps in ways we don't know or lead to overfitting.

When 49 % of variation is explained, simply, by the data employed (survey, publication, or submission, see p. 5 of manuscript), it makes me wonder how much is simply explained by differences in research design or modeling strategy.

4) Another concern is the conclusions drawn from the differences in meta-analytic effect drawn from figure 2. To me, the arguments of survey estimates being more "true" to the actual effect, because they are closer to the real-time events (p. 17) could easily be turned on its head.

Those estimates are also self-reported, and only a snapshot at one point.

a. People are bad at judging changes.

b. Productivity may have been severely hampered early on, when surveys were fielded, and then normalised over time.

c. Survey estimates are around 4.2 and 4.9 times larger than estimates from publication data (0.193/0.046) and submission data (0.193/0.039) (p. 7), but also has the smallest sample sizes (I think from looking at 1/SE in figure 2). Couldn't this easily be an effect of higher degrees of noise and lower power in this data?

d. Often, large-scale international surveys of researcher suffer from very low response rates and often mostly cover few countries (North American and Western Europe). At best, it makes it hard to generalise beyond these areas, and at worst this bias the results.

I appreciate a discussion of why survey estimates differ so much from publication- or submission-based studies, but this discussion is currently one-sided. The authors should consider if perhaps survey data suffers from other problems that could inflate estimates.

5) The authors seem to conflate gender inequality with gender bias throughout the paper (e.g. on line 55 or 281). In science studies, one would often denote a difference in outcome as a bias only if there is a provable mechanism of discrimination or differential treatment of women in e.g. journal accept rates, grant funding, etc. Instead, an indirect or structural difference could be a disparity or inequality brought on by other factors (e.g. women being disproportionally allotted more teaching load, domestic work, not being promoted to equal degree, etc.).

a. There is a great working paper by Traag and Waltman with a discussion of these, and I suggest the authors use some of their clarification (doi.org/10.48550/arXiv.2207.13665).

b. I think this is important, not because I want to downplay potential bias, but because it matters for possible interventions. What we should do to ameliorate unfair differential productivity differs according to the causal mechanism at work.

6) How was "before " and "during" pandemic defined? – this is not specified in the inclusion criteria, nor in how the data was extracted.

Lines 456-457: unclear how you define "before the pandemic" and form where you get the data on the numbers. E.g., my first impression was that you analysed yourself all publications for a given discipline as your baseline. Please clarify.

7) Screening

– Study screening section: the inclusion criteria are too vague to be reproducible. It is also incomplete. Was publication status used to exclude evidence? How about publication language and full text availability? Was there a limitation on the geographical scope or targeted career stages for the included studies and was this information coded in any way?

– Line 360: why not all articles were double-screened to avoid errors? It would not add much work.

– Line 368 and 382: no details provided how full-text articles were screened, e.g., in terms of the used software, also it is not specified if they were double-screened.

– Table 1 header: first and only mention that articles were screened from 2020 only – this is a first mention on having such a limit on publication searches – this should be in the search strategy description.

– The authors should describe whether some quality weighting was used or quality assessment played a part in the inclusion criteria.

8) Data and meta-data

– Extracting variables section: please refer the reader to a proper meta-data SI file describing in detail all extracted variables, at a level that enables replication of the data extraction and fully informed data reuse (the "meta-data" file archived on Zenodo does not provide such information in sufficient detail and some of the variables in the provided raw data file are quite cryptic).

– A data file is needed including a list of articles at the full-text screening stage, with individual reasons for exclusion.

– A meta-data file is needed describing in detail all extracted variables, at a level that enables replication of the data extraction and fully informed data reuse (the "meta-data" file archived on Zenodo does not provide such information in sufficient detail and some of the variables in the provided raw data file are quite cryptic).

9) Lines 324-327: Only 17 out of 21 eligible articles were included in the extracted data set – what happened with the remaining 4? Why were they not included in the dataset? From the data files available on Zenodo I can see these were not published as journals articles, however inclusion criteria do not state that only grey literature should be excluded.

10) Line 339-352: The search string used had only 14/17 sensitivity (82%). However, the next step of search strategy refined did not improve the sensitivity and just removed some of the irrelevant studies to reduce the screening effort.

Given the limited sample size, and limited scope of the searches, it is of concern that a significant proportion of potentially available evidence has been missed (at least 20%). The main reason for using "test set" (benchmarking) articles is to improve search sensitivity, with search precision being less important. Since it is hard to get 100% sensitivity in database searches, additional search strategies should be used to locate potentially missed articles, for example by collecting and screening articles that were cited by or cited already included articles (after the main database searching and screening is completed). Also, grey literature should have been searched.

11) Lines 350-251: the recommended 10% hit rate should be calculated from pilot screening round not from how many "test set" articles have been captured by the search string.

12) Lines 411-413: regression models per se are not effect sizes, please specify which estimates from these models were used as effect sizes and how they were combined with other effect sizes.

13) Analyses section: no description of any sensitivity analyses (e.g., doing analyses leaving out outliers or leaving-one-study-out approach).

14) Limitations section: there are many potential limitations of this study that are not mentioned here. For example, limitations of the search conducted to find the evidence – not searching grey literature, no other additional searches, not searching in languages other than English. This potentially resulted in biased evidence as well as limited the sample size. Also, some statistical analyses are less reliable due to small numbers of effect sizes within some of the levels of categorical variables tested.

15) Table S4 item 20d: Figures 2, 3, 5 show meta-regression results testing secondary hypotheses. Those are not sensitivity analyses.

---

## [Author Response]

Essential revisions1) I am concerned about the use of the included studies for a formal meta-analysis. Firstly, it is clear from the authors well-structured description in the methods section that many studies are not measuring the same outcome.Many studies use number of submitted or published articles, which may be a fairly comparable metric, but many of the survey studies use very different measures of productivity:a. Research time (N = 4), job-loss (N = 1), burnout (N = 1), and number of projects (N = 1).b. These are highly different outcomes, and should not be analysed as comparable.

We conducted a sensitivity analysis excluding the survey studies and found little change in the results. We now mention the sensitivity analysis in the main text:

“Our results changed little when we conducted a sensitivity analysis that excluded seven effect sizes using four measures of productivity from survey-based studies that are less directly comparable: research time (N = 4), job-loss (N = 1), burnout (N = 1) and the number of projects (N = 1). When excluding these effect sizes, the overall estimate is -0.063, 95% CI = [-0.0892- -0.0372] , SE = 0.0133, p < 0.001 and the estimate for survey studies only is -0.239, 95% CI = [-0.3419- -0.1352], SE = 0.0527, p < 0.001.”

We include all the studies in the main analysis because although they are not directly the same as productivity output in terms of submissions and publications, they are relevant to an academic’s productivity and likely reflect the overall change in the gender bias in academic productivity that our article aims to assess.

2) Secondly, it is not clear how the studies differ in their use of the pandemic as an "intervention".a. Some studies will ask respondents to compare their productivity before and after, while some studies will provide actual comparisons of productivity before and after. Both methods will have their share of drawbacks, but I do not think they are comparable.

All but 5 survey studies are about perceived impacts and not actual impacts, so these effects become separated in prediction 1b when we investigate survey and submission and publication studies as moderators. In addition, we recognize that none of our studies measure actual comparisons of individual-level productivity before and after the start of the pandemic. As we mention in the discussion, in the first submission, the findings of the studies comparing publication patterns:

“might reflect lower submission and acceptance rates of articles by women compared to their male colleagues or an increased drop-out of women from academia caused by the pandemic.”

b. The different survey studies will also have been fielded at different times, making the "intervention" (the pandemic) very different. Perhaps the loss of research time is most acute in the beginning, before academics have had time to adapt? Or later, because of burn-out?

We now include this point in the ‘Limitations’ section:

“We recognise there are differences between article studies in the length of time sampled before the pandemic (mean = 11 months, standard deviation = 10 months, range = 1-50 months) and during the pandemic (mean = 7 months, standard deviation = 5 months, range = 1-17 months). Survey studies were fielded at different times, (mean = 21/08/2020, standard deviation = 99 days, range = 20/04/2020 – 28/02/21) which potentially affects participants’ feelings of productivity changes.”

In addition, we mention in the discussion:

“It is likely that the article studies we could include in our study underestimates the long-term effects of the pandemic, which might span over many years.”

3) Thirdly, the diversity of research design and modelling techniques makes it hard to conduct formal comparisons of effect estimates.

Diversity in research designs is common in most fields as there are generally no set of standards on how to conduct a certain type of a study. This is why there is commonly a very large difference in between study heterogeneity in effect sizes. However, a part of the heterogeneity will be explained by certain common features of the studies, and we have tried to capture these in our analysis.

None of the included studies are randomised controlled experiments. In fact, the research design (and modelling techniques) differs immensely across included studies. Therefore:a. We do not know to what degree individual, and thus by extension, individual gender gap estimates can be causally interpreted. This should be more explicitly acknowledged in the article.

We now include the following sentence in the ‘Limitations’ of the ‘Methods’:

“The patterns we describe should be seen as a potential indication that biases exist, but alternative approaches are needed to speculate about potential underlying causes and remedies.”

Our finding that the gender gap has increased in academic productivity, is contextualised around evidence in other studies that the pandemic has enhanced gendered work roles and disadvantaging women. We start the final paragraph of the discussion with:

“Our study has several limitations, which are outlined in the Limitations of the methods and summarised here. Causes of the increased gender disparity in research productivity during the pandemic are not identified.”

b. Different estimates may be relatively closer to a causal effect, depending on the research design. By treating all designs equally, the authors gloss over these crucial differences in what we can gleam from each study.

Unlike medical meta-analyses, our approach does not assume that there is one causal effect that is being estimated by all studies. Rather, we assume that there can be multiple effects that all shape the gender inequality in academic productivity. Our question is to assess whether overall, these effects have changed during the pandemic such that also the gender inequality has either generally increased or decreased. The summary statements we provide reflect the average effect rather than our estimate of the true effect. We now clarify this in the introduction:

“We assume that the pandemic might have influenced the multiple aspects that jointly affect gender inequality in research productivity, and our estimate reflects whether on average these effects have increased or decreased gender inequality.”

and in the discussion:

“Future studies might therefore detect different effects of the pandemic on gender disparity in productivity, depending on the outcomes they assess and the time-frame over which these changes are analysed.”

and:

“Despite the heterogeneity in our sample, with regards to different outcome measures reflecting research productivity and different approaches to estimate effects, we overall find that the majority of studies performed to date indicate that the gender gap in research productivity has increased during the pandemic. This finding suggests that the COVID-19 pandemic has likely influenced many of the processes that contribute to differences in the achieved research productivity of women compared to men.”

The most fundamental difference in research design of studies is between studies that assess productivity by numbers of male and female authors before and during the pandemic, and those that use surveys to assess productivity differences of individuals. We separate these studies in prediction 1b and Figure 2, and then subsequently only use measures of numbers of submissions and publications in downstream analyses investigating effects of research field, size of previous gender gap and authorship position. Finally, while Risk of Bias is commonly a part of meta-analysis in medicine, and sometimes in environmental evidence, it is almost never done in other fields. While we agree that this is an important component of meta-analysis, unfortunately there are no Risk of Bias standards or tools for the kind of studies we have used.

c. It is not clear what "estimand" we get from the meta-analytical model. Some studies are cross-sectional, and will give us the average between-subject difference in productivity. Other studies, longitudinal ones, may give us the "difference in differences", i.e. the average difference in within-subject publication productivity trend. These are fundamentally two different answers to the question: Has the pandemic exacerbated the gender gap in research?

We referred to the difference between cross-sectional and longitudinal studies in our discussion, acknowledging that for cross-sectional studies selective disappearance may occur:

“Such a change might reflect lower submission and acceptance rates of articles by women compared to their male colleagues or an increased drop-out of women from academia caused by the pandemic.”

All the included studies investigating productivity before and during the pandemic were cross-sectional. The ‘publishing’ studies aggregate data across individuals, and the surveys were only conducted at a single time point, asking people whether they thought the pandemic had affected their productivity. We also clarify in the ‘Extracting variables’ sub-section on ‘Effect sizes’ that:

“Six studies investigated numbers of articles at different time points during the course of the pandemic. From these studies, we calculated 30 effect sizes using numbers of articles across the entire pandemic period.”

d. The guidance from e.g. Cochrane is to use a random effects model to account for variation in what effect is estimated. The authors seem to use a random effects model to account for dependency between effects extracted, but I think they should also be more explicit in leveraging this for accounting for variation in what effect is estimated.

We are limited in using such an approach, because, as also pointed out by the reviewers, given the diverse factors shaping academic productivity we can only estimate the potential average effect of the pandemic on gender biases in research productivity rather than one true effect.

e. Each included paper will have different modelling strategies. E.g. the Myers et al. 2020 paper uses regularised regression (Lasso), while the Andersen et al. (2020) used a hierarchical logit model. These differences may contribute to differences in estimated gender gaps in ways we don't know or lead to overfitting.When 49 % of variation is explained, simply, by the data employed (survey, publication, or submission, see p. 5 of manuscript), it makes me wonder how much is simply explained by differences in research design or modeling strategy.

We include the following in the ‘Limitations’ sections of the ‘Methods’:

“We used raw data to calculate effect sizes using the same modelling techniques wherever possible. This was not possible in 10 studies, where we consequently used effect size as provided in the study. We recognize that their different modelling techniques may have contributed to the estimated effect sizes.”

4) Another concern is the conclusions drawn from the differences in meta-analytic effect drawn from figure 2. To me, the arguments of survey estimates being more "true" to the actual effect, because they are closer to the real-time events (p. 17) could easily be turned on its head.Those estimates are also self-reported, and only a snapshot at one point.a. People are bad at judging changes.

We include the following at the end of the second paragraph of the Discussion:

“As with any survey study, survey data are prone to subjectivity and suffer from limited numbers of respondents sampled which may influence the overall estimate.”

b. Productivity may have been severely hampered early on, when surveys were fielded, and then normalised over time.

In our data, surveys were not fielded earlier (or later) compared to when the data collection for the publication studies occurred (both occurred at a median of 6-7 months into the pandemic). In addition, the reports we had found mostly indicated that research projects take longer between conception and final publication than the duration of the pandemic, suggesting that the effect of the pandemic on publication rates might persist for extended periods of time. However, the patterns over time are indeed difficult to predict, and we have therefore amended our statement in the discussion:

“It is likely that the article studies we could include in our study underestimate the long-term effects of the pandemic, which might span over many years.. However, the exact time dynamics are difficult to predict as adjustments and changes in conditions might lead to a normalising in production patterns over time (Clark 2023).”

c. Survey estimates are around 4.2 and 4.9 times larger than estimates from publication data (0.193/0.046) and submission data (0.193/0.039) (p. 7), but also has the smallest sample sizes (I think from looking at 1/SE in figure 2). Couldn't this easily be an effect of higher degrees of noise and lower power in this data?

We include a leave-one-out sensitivity analysis for survey studies only. We did not detect evidence of publication bias that might have substantially shifted the values up, but with the small sample there could be relatively high uncertainty around the mean effect estimate of the survey studies. The results of this are described:

“Overly large effect sizes did not change the results in a leave-one-out analysis, which repeatedly fitted the overall model as in prediction 1a, but for survey studies only, leaving out one effect size at a time to see the effect on the overall survey-study estimate (Figure S2). Leaving out the most influential effect size, the overall estimate was -0.183, 95% CI = [-0.270 - -0.096] , SE = 0.045, p < 0.001.”

and methods outlined:

“We also performed a leave-one-out analysis using the ‘leave1out’ function in ‘metafor’. This was a meta-analysis on survey studies, leaving out exactly one study at a time to see the effect of individual studies on the overall survey-study estimate.”

d. Often, large-scale international surveys of researcher suffer from very low response rates and often mostly cover few countries (North American and Western Europe). At best, it makes it hard to generalise beyond these areas, and at worst this bias the results.

We include the following in the ‘Limitations’ section of the ‘Methods’:

“Conclusions from survey studies are also limited to North American and Western European, since 18/23 studies are exclusive to or dominated by respondents from these regions.”

I appreciate a discussion of why survey estimates differ so much from publication- or submission-based studies, but this discussion is currently one-sided. The authors should consider if perhaps survey data suffers from other problems that could inflate estimates.

We have balanced the discussion for survey data following Comment 4 as in the preceding comments and corresponding author action/responses.

5) The authors seem to conflate gender inequality with gender bias throughout the paper (e.g. on line 55 or 281). In science studies, one would often denote a difference in outcome as a bias only if there is a provable mechanism of discrimination or differential treatment of women in e.g. journal accept rates, grant funding, etc. Instead, an indirect or structural difference could be a disparity or inequality brought on by other factors (e.g. women being disproportionally allotted more teaching load, domestic work, not being promoted to equal degree, etc.).a. There is a great working paper by Traag and Waltman with a discussion of these, and I suggest the authors use some of their clarification (doi.org/10.48550/arXiv.2207.13665).

We have revisited where we express “bias” and now avoid use of the word unless in context of mechanisms at play.

b. I think this is important, not because I want to downplay potential bias, but because it matters for possible interventions. What we should do to ameliorate unfair differential productivity differs according to the causal mechanism at work.

We have added a sentence to our ‘Conclusions’:

“Our study cannot speak to the potential mechanisms that might have led to changes in gender inequality during the pandemic and is therefore limited in deriving suggestions for potential interventions to potentially ameliorate unfair differential productivity besides indicating that such inequalities appear pervasive.”

6) How was "before " and "during" pandemic defined? – this is not specified in the inclusion criteria, nor in how the data was extracted.Lines 456-457: unclear how you define "before the pandemic" and form where you get the data on the numbers. E.g., my first impression was that you analysed yourself all publications for a given discipline as your baseline. Please clarify.

We specify in the ‘Methods’ for inclusion criteria for the full-text screen that time periods for before and after the pandemic are chosen according to the authors’ discretion:

“Full texts were then screened to identify articles that had included four qualifiers: (1) both genders, (2) some quantifiable metric of academic productivity measured, and (3) compared a time period before the pandemic with a time period during the pandemic (time periods chosen according to the authors’ discretion), (4) primary data.”

We remind readers of this in the ‘Methods’:

“Effect size: We extracted values needed to calculate 130 effect sizes from 55 articles investigating the effect of the pandemic on academic research productivity of both genders, comparing the productivity before and during the pandemic, using time periods chosen according to authors’ discretion”

and:

“Previous gender disparity: For the article studies with available data (N=99 effect sizes), we recorded the number of female and male authors before the pandemic, as defined by the time-period sampled in the respective study and use this ratio as a proxy for size of the previous gender disparity in that population sampled.”

7) Screening– Study screening section: the inclusion criteria are too vague to be reproducible. It is also incomplete? Was publication status used to exclude evidence? How about publication language and full text availability?

We include the following in the ‘Study screening’ section of the ‘Methods’:

“We included any text returned by the database, of any publication status, including grey literature in any language, though all texts had abstracts with an English version.”

We include in the ‘Iterating the search’ of the ‘Methods’:

“Two studies with full texts in Spanish were translated to English by Google translate, of which one was included.”

We include in the ‘Iterating the search’ section of the ‘Methods’:

“All full-texts were available”.

We include the following in the ‘Extracting variables’ sub-section of the ‘Methods’:

“We also extracted data for the following variables, to enable description of the datasets: date range before the pandemic; date range during the pandemic; geographic region; data availability for gender and geographic region interaction effect; data availability for gender and career stage/age interaction effect; data availability for gender and parent status interaction effect; gender assignment accuracy threshold for article studies and gender inference method used (Supplementary file 2, see also ‘Limitations’).”

Was there a limitation on the geographical scope or targeted career stages for the included studies and was this information coded in any way?

We acknowledge in the ‘Limitations’ section of the ‘Methods’:

“Most studies explored academic populations worldwide (N = 99 effect sizes), or from Western (N = 28 effect sizes) regions, but not the Global South (N = 3 effect sizes) limiting investigation of interaction effects between geographic regions. Although 22/130 effect sizes from 8/55 studies held data subdivided between geographic regions, we did not extract separate effect sizes as they differed in the scale of geographic region sampled, which limited our ability to make geographic comparisons. Conclusions from survey studies are also limited to North American and Western European, since 18/23 studies are exclusive to, or have the majority of respondents from these regions. Additionally, we recognise there are differences between article studies in the length of time considered as before the pandemic (mean = 11 months, standard deviation = 10 months, range = 1-50 months) and during the pandemic (mean = 6.6 months, standard deviation = 5 months, range = 1-17 months). Survey studies were fielded at different times, (mean = 21/08/2020, standard deviation = 99 days, range = 20/04/2020 – 28/02/21) which potentially affects participants’ beliefs of productivity changes”

– Line 360: why not all articles were double-screened to avoid errors? It would not add much work.

We include the following the ‘Study screening’ section of the ‘Methods’:

“To reduce workload, we did not double-initial-screen 160 texts because of 97% agreement rates achieved.. We acknowledge this may miss up to 13% of studies (Waffenschmidt et al., 2019)”

– Line 368 and 382: no details provided how full-text articles were screened, e.g., in terms of the used software, also it is not specified if they were double-screened.

We include in the ‘Study screening’ section of the ‘Methods’:

“Full texts were then screened to identify articles that had included four qualifiers: (1) both genders, (2) some quantifiable metric of academic productivity measured, and (3) compared a time period before the pandemic with a time period during the pandemic (time periods chosen according to the authors’ discretion), (4) primary data. Full-text screening was conducted by recording which, if any of the four qualifiers were missing for each text in an excel spreadsheet and only performed by K.L.”

– Table 1 header: first and only mention that articles were screened from 2020 only – this is a first mention on having such a limit on publication searches – this should be in the search strategy description.

We include in the ‘Search process’ section of the ‘Methods’:

“All searches were filtered for texts from 2020 onwards.”

– The authors should describe whether some quality weighting was used or quality assessment played a part in the inclusion criteria.

We include in the ‘Study screening’ section of the ‘Methods’:

“Studies were not assigned quality weightings as there is no common standard to do this.”

8) Data and meta-data– Extracting variables section: please refer the reader to a proper meta-data SI file describing in detail all extracted variables, at a level that enables replication of the data extraction and fully informed data reuse (the "meta-data" file archived on Zenodo does not provide such information in sufficient detail and some of the variables in the provided raw data file are quite cryptic).

We include a datafile (Supplementary file 2, “Variables” sheet) describing all variables extracted. We include the following in the ‘Extracting variables’ section of ‘Methods’:

“We also extracted data for the following variables to enable description of our datasets: date range before the pandemic; date range during the pandemic; geographic region; data availability for gender and geographic region interaction effect; data availability for gender and career stage/age interaction effect; data availability for gender and parent status interaction effect; gender assignment accuracy threshold for article studies and gender inference method used. Please see Supplementary file 2, “Variables” sheet, for descriptions of all the variables”

– A data file is needed including a list of articles at the full-text screening stage, with individual reasons for exclusion.

We include a datafile (Supplementary file 1) describing individual reasons for excluding articles.

We included in the ‘Iterating the search’ section of the ‘Methods’:

“For these 169, we screened the full texts, excluding 120 studies (53 studies missed one qualifier, 38 studies missed two qualifiers, 27 studies miss three qualifiers, 1 study missed four qualifiers) and keeping 50 (including the 25 identified in the original search) that all contained the necessary information to calculate the effect sizes (Supplementary file 1)”

– A meta-data file is needed describing in detail all extracted variables, at a level that enables replication of the data extraction and fully informed data reuse (the "meta-data" file archived on Zenodo does not provide such information in sufficient detail and some of the variables in the provided raw data file are quite cryptic).9) Lines 324-327: Only 17 out of 21 eligible articles were included in the extracted data set – what happened with the remaining 4? Why were they not included in the dataset? From the data files available on Zenodo I can see these were not published as journals articles, however inclusion criteria do not state that only grey literature should be excluded.

There are five, not four studies, because the original submission listed 14/17 scoped articles indexed in Scopus picked up by the search, when this should have been 13/17 articles. This has since been corrected.

We have included the five articles that were scoped but not picked up in the databases and re-run the analyses using these data and state this in the Iterating the search section of the ‘Methods’:

“Five studies found in the scoping search but not returned in database searches were also included, resulting in 55 total studies used to extract variables for analysis.”

10) Line 339-352: The search string used had only 14/17 sensitivity (82%). However, the next step of search strategy refined did not improve the sensitivity and just removed some of the irrelevant studies to reduce the screening effort.Given the limited sample size, and limited scope of the searches, it is of concern that a significant proportion of potentially available evidence has been missed (at least 20%). The main reason for using "test set" (benchmarking) articles is to improve search sensitivity, with search precision being less important. Since it is hard to get 100% sensitivity in database searches, additional search strategies should be used to locate potentially missed articles, for example by collecting and screening articles that were cited by or cited already included articles (after the main database searching and screening is completed). Also, grey literature should have been searched.

Minimising work load at the expense of potentially missing some available evidence was a trade-off we were willing to make given the need to quickly produce results in the context of a potentially pressing issue. For this, we chose to use litsearchR to improve efficiency of finding texts and to not perform forwards and backwards searches. Additionally, given the research topic is so new, we will consistently be finding new articles if we performed backwards and forwards searches and so we chose to not perform these. We did not search specific grey literature databases, but did not filter against grey literature during screening. We welcome future research to incorporate our scripts and methods to monitor this new research topic including more research as and when it becomes available.

11) Lines 350-251: the recommended 10% hit rate should be calculated from pilot screening round not from how many "test set" articles have been captured by the search string.

We have deleted the statement about “above the 10% hit rate recommended by Foo et. al”.

12) Lines 411-413: regression models per se are not effect sizes, please specify which estimates from these models were used as effect sizes and how they were combined with other effect sizes.

We include in the ‘Extracting variables’ section of the ‘Methods’:

“For 10 effect sizes which did not have data available to calculate our own effect sizes, we used already calculated percentage changes in the gender gap in academic productivity as predicted from lasso regression (N = 2), Somers’ δ (N = 2), ordered logistic regression (N = 1) and mixed-effect models (N = 5).”

13) Analyses section: no description of any sensitivity analyses (e.g., doing analyses leaving out outliers or leaving-one-study-out approach).

We include the following sensitivity analysis leaving studies in the ‘Are our results robust?’ section of the ‘Results’:

“Our results show little change in a sensitivity analysis that excludes seven effect sizes using four measures of productivity from survey-based studies that are less directly comparable: research time (N = 4), job-loss (N = 1), burnout (N = 1) and the number of projects (N = 1). When excluding these effect sizes, the overall estimate is -0.063, 95% CI = [-0.0892- -0.0372] , SE = 0.0133, p < 0.001 and the estimate for survey studies only is -0.239, 95% CI = [-0.3419- -0.1352], SE = 0.0527, p < 0.001.”

We now include a leave-one-out analysis for the overall effect of the pandemic on the gender gap and record the results in the ‘Are our results robust?’ section of the ‘Results’:

“Overly large effect sizes did not change the results in a leave-one-out analysis, which repeatedly fitted the overall model as in prediction 1a, but for survey studies only, leaving out one effect size at a time to see the effect on the overall survey-study estimate (Figure S2). Leaving out the most influential effect size, the overall estimate was -0.183, 95% CI = [-0.270 - -0.096] , SE = 0.045, p < 0.001.” and display this as a forest plot as Figure S2.

These analyses are written in ‘Analyses’ section of the ‘Methods’:

“We applied a sensitivity analysis testing prediction 1a (overall pandemic effect on gender gap) and prediction 1b (method of research productivity influence on pandemic effect on gender gap) excluding seven effect sizes using four measures of productivity from survey-based studies that are less directly comparable: research time (N = 4), job-loss (N = 1), burnout (N = 1) and the number of projects (N = 1). We also performed a leave-one-out analysis using the ‘leave1out’ function in ‘metafor’. This performed a meta-analysis on survey studies, leaving out exactly one study at a time to see the effect of individual studies on the overall estimate for survey studies.”

14) Limitations section: there are many potential limitations of this study that are not mentioned here. For example, limitations of the search conducted to find the evidence – not searching grey literature, no other additional searches, not searching in languages other than English. This potentially resulted in biased evidence as well as limited the sample size. Also, some statistical analyses are less reliable due to small numbers of effect sizes within some of the levels of categorical variables tested.

We have considerably expanded our ‘Limitations’ section in the ‘Methods’, including all the examples given by the reviewer:

“Our focus is on comparing the effect of the pandemic on women relative to men. We recognize that gender extends beyond this comparison, and that biases are even more likely to target individuals whose identities are less represented and often ignored. These biases also reflect in a lack of studies of the full diversity of gender. While several of the surveys we include had the option for respondents to identify beyond the binary women/men, none of these studies report on these individuals, presumably because of the respective small samples. In addition, all studies using numbers of submissions or publications (38 out of 55) to measure research productivity used automatic approaches that are more likely to mis-gender individuals as they inferred binary gender based on first names. While these approaches seemingly offer the potential to identify trends in larger samples, they themselves introduce and reinforce biases in relation to gender that are hard to assess, intersecting with biases in ethnicity as these approaches are often restricted to names common in English speaking countries (Mihaljević et al., 2019). We do not include grey literature databases in our searches, which may bias our samples to studies with positive effects. We did not perform forwards or backwards searches, meaning we may have missed some relevant studies. However, we expect the literature on the topic to grow, and hope that further work will build on our study and add these new effect sizes to our dataset. Most studies explored academic populations worldwide (N = 99 effect sizes), or from Western (N = 28 effect sizes), regions, but not the Global South (N = 3 effect sizes), limiting investigation of interaction effects between geographic regions. Although 22/130 effect sizes from 8/55 studies held data subdivided between geographic regions, we did not extract separate effect sizes as they differed in the scale of geographic region sampled, which limited our ability to make geographic comparisons. Conclusions from survey studies are also limited to North American and Western European, since 18/23 studies are exclusive to or have the majority of respondents from these regions. We recognise there are differences between article studies in the length of time considered as before the pandemic (mean = 11 months, standard deviation = 10 months, range = 1-50 months) and during the pandemic (mean = 7 months, standard deviation = 5 months, range = 1-17 months). Survey studies were fielded at different times, (mean = 21/08/2020, standard deviation = 99 days, range = 20/04/2020 – 28/02/21) which potentially affects participants’ beliefs of productivity changes. Investigating research field and authorship position effects is limited by the unequal and sometimes small sample sizes of variables that are compared. We use 10 studies that provide effect sizes without raw data available to calculate our own effect size and recognise their differing modelling techniques may contribute to the estimated effect sizes. The patterns we describe should be seen as a potential indication that biases exist, but alternative approaches are needed to speculate about potential underlying causes and remedies.”

We also close the ‘Discussion’ with a summary of limitations:

“Our study has several limitations, which are outlined in the Limitations section of the methods and summarised here. Causes of the increased gender disparity in research productivity during the pandemic are not identified. Gender is investigated only as a binary variable, mostly using first-name prediction tools, with no investigation into non-binary or transgender. Geographic regions are not investigated, limiting generalisability as our samples are not equally representative of all geographic regions. All studies are to some extent unique by combined differences in sampling method and analyses performed.”

15) Table S4 item 20d: Figures 2, 3, 5 show meta-regression results testing secondary hypotheses. Those are not sensitivity analyses.

We changed the entry for 20d sensitivity analyses in Table S4 to refer to the places where we added the results from the sensitivity analyses in the ‘Are our results robust?’ section of the methods: “Page 15, Figure S2”.